# Fabrication of ZnO@Plant Polyphenols/Cellulose as Active Food Packaging and Its Enhanced Antibacterial Activity

**DOI:** 10.3390/ijms23095218

**Published:** 2022-05-07

**Authors:** Jingheng Nie, Ziyang Wu, Bo Pang, Yuanru Guo, Shujun Li, Qingjiang Pan

**Affiliations:** 1Key Laboratory of Bio-Based Material Science & Technology (Ministry of Education), College of Material Science and Engineering, Northeast Forestry University, Harbin 150040, China; njh1024856068@163.com (J.N.); wuziyang@nefu.edu.cn (Z.W.); 18845893226@163.com (B.P.); lishujun@nefu.edu.cn (S.L.); 2Key Laboratory of Functional Inorganic Material Chemistry (Ministry of Education), School of Chemistry and Materials Science, Heilongjiang University, Harbin 150080, China

**Keywords:** plant polyphenol, ZnO, cellulose, antibacterial activity, package material

## Abstract

To investigate the efficient use of bioresources and bioproducts, plant polyphenol (PPL) was extracted from larch bark and further applied to prepare ZnO@PPL/Cel with cellulose to examine its potential as an active package material. The structure and morphology were fully characterized by XRD, SEM, FTIR, XPS and Raman spectra. It was found that PPL is able to cover ZnO and form a coating layer. In addition, PPL cross-links with cellulose and makes ZnO distribute evenly on the cellulose fibers. Coating with PPL creates a pinecone-like morphology in ZnO, which is constructed by subunits of 50 nm ZnO slices. The interactions among ZnO, PPL and cellulose have been attributed to hydrogen bonding, which plays an important role in guiding the formation of composites. The antibacterial properties against Gram-negative *Escherichia coli* (*E. coli*) and Gram-positive *Staphylococcus aureus* (*S. aureus*) were tested by the inhibition zone method. Our composite ZnO@PPL/Cel has superior antibacterial activity compared to ZnO/Cel. The antibacterial mechanism has also been elaborated on. The low cost, simple preparation method and good performance of ZnO@PPL/Cel suggest the potential for it to be applied as active food packaging.

## 1. Introduction

With industrial development and social progress, fossil resources have become very important as raw materials for industry. Massive use of fossil resources results in the emission of carbon dioxide, which has become a serious problem for environmental and sustainable development. To reduce the production of carbon dioxide and achieve carbon neutrality, substitute resources and technology need to be developed [1,2]. Using bioresources with high added-value is an effective way to fulfill this aim [3,4]. Thus, bioresources, produced by adsorption of CO_2_, have been considered as ideal substitute raw materials for avoiding the use of fossil resources. Plant polyphenols (PPL), the fourth most abundant bioresource in nature other than cellulose, lignin and hemicellulose, is widespread throughout all kinds of plants. Many components of PPL, such as catechin, flavones, anthocyan, and phenolic acid, have been determined and their structures depend upon the flavan-3-ol starter and extension units. Due to the diversity of functional groups in PPL, it is believed to have beneficial health effects for humans and work as anticancer agent [5,6,7,8]. In addition to its anticancer properties, PPL has antioxidant, anti-carcinogenic, cardioprotective, antimicrobial and neuro-protective activities [9,10,11,12].

Due to structural complexity of PPL, there are many kinds of PPL and all of them have the same character: an abundance of hydrogen groups in their units. As a result of this structure, PPL can release H^+^ ions from their hydroxyl groups and delocalize free electrons, which can kill bacteria.

Though PPL has many fascinating merits, studies on PPL are limited compared with those on cellulose, lignin or hemicellulose. The reason for this is the difficulty of its extraction and separation from plants. As all PPLs have similar structures and properties, it is complex and expensive to obtain specific components from the PPL mixture. Thus, finding a new way to use PPL mixture efficiently is an ongoing challenge.

As an innovative and promising technology, active food packaging would prevent food from oxidation and pathogen growth [13,14,15,16]. Food would keep more safely during transportation, storage and sale if packaged in these kinds of materials. To achieve this character, functional materials, such as antioxidants and/or antimicrobial agents, are embedded and loaded in the package material. The popular antibacterial materials are metallic materials, non-metallic composites [17,18,19,20,21,22], such as nano-silver, ZnO, and MgO [23,24,25]. However, expensive metal materials containing silver are not suitable for mass use [26,27,28,29,30]. ZnO, as an environmentally friendly, nontoxic and cheap semiconductor, is widely studied for its high antibacterial activity [31]. According to previous reports, different preparation methods influence ZnO band structure and then result in differences in antibacterial properties [32,33]. Thus, a variety of methods are applied to fabricate ZnO, such as doping hetero-atoms or coupling with other materials [34,35]. There are some studies that combine ZnO with cellulose and study the antibacterial properties of the resultant composites. It has been found that ZnO can form interfacial bonds with cellulose and endow the composite with good antibacterial properties [36]. Different sources of cellulose, such as nanocellulose, bacterial cellulose and cellulose from pulp [37,38,39], were adopted to fabricate the composite, and antibacterial property against *Salmonella typhi*, *Staphylococcus aureus*, *Escherichia coli* and *Staphylococcus aureus* were fully investigated. However, reports about controlling synthesis of ZnO with small size and plenty of defects, which play an important role in improving its performance, are still rare. Thus, new technology to prepare ZnO with high antibacterial activity is still in demand.

In this work, we prepared an active food package material of ZnO@Plant polyphenols/cellulose (ZnO@PPL/Cel) via a hydrothermal method. The PPL were extracted from the byproduct material of larch bark and used as a mixture without separation. This method would greatly reduce of preparation costs of ZnO@PPL/Cel. In ZnO@PPL/Cel composite, PPL is found coated on ZnO via hydrogen bonding but it can also build interfacial bonding with hydroxyl groups of cellulose and construct a composite package material. However, the coating of PPL restricts the growth of ZnO and creates many defects on these nanoparticles, which can enhance the antibacterial activity. The low cost, lack of secondary pollution and benefits to human health suggests the potential application of this material as active food packaging. The synthesis route and connection of components is shown in Figure 1.

## 2. Materials and Methods

### 2.1. Reagents

Anhydrous ethanol was purchased from Tianjin Fuyu Fine Chemical Co., Ltd. (Tianjing, China). Zinc acetate was purchased from Tianjin Comiou Chemical Reagent Co., Ltd. (Tianjing, China). Cellulose was purchased from Hangzhou Fuyang Beimu Pulp and Paper Co., Ltd. (Hangzhou, China). Agar, beef extract and peptone were purchased from Fujian Shishi Gaoxin Agar Food Co. (Fujian, China) and Beijing Oboxing Biotechnology Co., Ltd. (Beijing, China). All the chemical reagents were analytically pure, and agar, beef extract and peptone are biological reagents.

### 2.2. Extraction of PPL

Larch bark polyphenols were extracted from larch bark powder by an organic solvent extraction method. A 20 g larch bark powder sample and 200 mL 50% ethanol solution were added to a 250 mL conical flask and shaken at 50 °C (the ratio of bark raw material to ethanol solution was 1 g:10 mL). After a 6 h extraction, the residue of larch bark was separated by filtration and the PPL solution was obtained. Each 100 mL PPL extraction contained about 1.863 g PPL.

### 2.3. Preparation of ZnO@PPL/Cel

Zinc acetate solution was obtained by dissolving 3.0 g zinc acetate in 50 mL of distilled water and then adding 1.0 g cellulose. After 1 h of stirring, 5 mL PPL solution was added and the pH of the mixture was adjusted to 11 with 2 mol L^−1^ NaOH solution. Stirring was continued for 0.5 h; then the mixture was transferred into a 100 mL hydrothermal reactor and heated at 100 °C for 10 h. After cooling, ZnO@PPL/Cel was obtained after washing and drying.

### 2.4. Antioxidant Experiments

Antioxidant activity of the prepared samples was tested by scavenging 2,2-diphenyl-1-picrylhydrazyl (DPPH) free radicals [40,41]. A 1.0 g sample was added to 25 mL of anhydrous ethanol and shaken for 24 h at room temperature in an oscillation instrument. The DPPH ethanol solution was prepared by dissolving 1.0 mg of DPPH in 20 mL of anhydrous ethanol solution (the volume ratio of anhydrous ethanol to deionized water was 4:1). Then a 4 mL sample was mixed with 4 mL of DPPH ethanol solution, and the absorbance of the mixture was measured by a UV spectrophotometer at 517 nm after 1 h dark reaction. The scavenging rate was calculated according to the following equation:$$DPPH\;scavenging\;activity\left(\%\right)=\frac{A_{0}-A}{A_{0}}\times 100\%$$
where *DPPH* scavenging activity denotes the free radical scavenging rate of *DPPH*, *A*_0_ is the absorbance of *DPPH* ethanol solution at 517 nm, and *A* stands for the absorbance of the mixture.

### 2.5. Antibacterial Experiments

*Staphylococcus aureus* (*S. aureas*) and *Escherichia coli* (*E. coli*) were used as reference strains for antimicrobial tests. The antibacterial effect of the composite was tested and evaluated by the inhibition zone method [42,43]. A 0.2 mL bacterial suspension (106 CFU/mL) was placed on the surface of the culture medium. A ZnO@PPL/Cel pellet with a diameter of 15 mm was placed in the middle of the surface of the culture medium. The culture dish was placed in a 37 °C incubator for 12 h. The growth of the bacteria was observed and the sizes of the inhibition zones were measured.

### 2.6. Water Vapor Transmission Rate (WVTR) Experiments

The water vapor transmission rate (WVTR) of samples was tested according to the reference [44]. In the experiment, ZnO@PPL/Cel was processed into a pellet with a diameter of 25 mm and thickness of 0.7 mm. Then the sample was placed into a container for 12 h (humidity 50%, and temperature 23 °C). The balanced sample was fixed at the mouth of the weighing bottle with 1.0 g anhydrous calcium chloride. Then the weighing bottle was placed in a sealed container at 23 °C with 90% relative humidity. The weight of bottle was measured every 2 h. The WVTR was calculated by the following equation:$$WVTR\;\left(\frac{g}{h\cdot m^{2}\cdot mm}\right)=\frac{W}{T\cdot S\cdot t}$$
where *W* (g) is the quantity difference between the certain intervals, *T* (mm) and *S* (m^2^) are the average thickness and surface area of the sample, respectively, and *t* (h) is the time interval.

### 2.7. Characterization

The crystalline structures of samples were analyzed by X-ray diffractometer (D/max-RB, Japan Scientific Instrument Co., Ltd., Tokyo, Japan). The incident light was set to CuKα, λ = 1.5418 Å, the scanning voltage was set to 40 kV, the scanning current was set to 30 mA, and the scanning rate was set to 5°/min. Fourier transform infrared spectrometer (FT-IR, PerkinElmer, Hopkinton, MA, USA) was used to analyze the functional groups of samples. The UV spectra were measured with a TU-1901 ultraviolet-visible spectrophotometer (Beijing Purkinje General Instrument Co., Ltd., Beijing, China). Laser confocal Raman spectroscopy of LabRAM HR800 (Jobin Yvon, Paris, France) was used to analyze the molecular vibrations. The microstructure of the samples was examined by scanning electron microscope (JSM-7500F, JEOL, Tokyo, Japan). The chemical environment around PerkinElmerC, O and Zn in the composites was characterized by X-ray photoelectron spectroscopy (XPS, Thermo Fisher Scientific Co., Ltd., Waltham, MA, USA).

## 3. Results and Discussion

### 3.1. Structure of ZnO@PPL/Cel

The structure of composite as revealed by the XRD results is shown in Figure 1a. From the Figure 1a, one can see that the PPL has a wide peak around 20.4°, which implies the amorphous nature of PPL. ZnO@PPL/Cel has similar diffraction peaks to ZnO and ZnO/Cel: diffraction peaks are found at 32.7°, 35.3°, 37.1°, 48.5°, 57.4°, 63.7°, 67.1°, 68.7°, and 69.8°, which are the characteristic diffractions of (100), (002), (101), (102), (110), (103), (200), (112), (201) of ZnO with hexagonal structure (PDF#36-1451). The sharpness of the diffraction peaks also indicates good crystallization of ZnO in ZnO@PPL/Cel. In addition, there is another peak at 2θ = 23.0° in both ZnO/Cel and ZnO@PPL/Cel, which is the diffraction peak of cellulose I type. Since PPL in the composite is in the amorphous phase, the diffraction intensity is too low to be observed.

According to the standard ZnO, the intensity of (002) is lower than the (100) plane. However, when PPL was involved during preparation, the (002) plane in ZnO@PPL/Cel becomes stronger and higher than that of (100). This result means that ZnO has a predominant growth orientation along the C-axis. ZnO is a polar crystal and has a Zn terminus on (001) planes [45], which is positively charged. PPL may prefer to coordinate with O in ZnO via hydrogen bonding on the other planes except for the (001) plane, which results in the preferential growth along the C axis and enhances the diffraction intensity of the (002) plane. This is evidence that PPL can couple with PPL and cellulose, which finally build the hierarchical structure of ZnO@PPL/Cel. Compared with ZnO/Cel, ZnO@PPL/Cel is more stable because the abundance of hydroxyl groups in both PPL and cellulose, which would strengthen the interaction between them.

Since the stability of PPL could be affected by alkaline conditions and the hydrothermal treatment, PPL before and after treatments was evaluated by the FT-IR and UV spectra. The sample treated by alkalinity of pH = 11 was named PPL-11, and it was further hydrothermally treated at 100 °C for 10 h and the resultant sample was named PPL-11-H. The UV spectra (Figure 1b) show the same character: all PPL samples have the adsorption peaks at 200 and 280 nm, which are related to π→π* and n→π* transitions, respectively. This also indicates the PPL before and after treatment has the same functional groups because of the complicated and diverse structures of PPL extraction. Thus, high-temperature and alkaline treatments can change the quantity of functional groups in PPL but not their types.

In the FT-IR spectra in Figure 1c, the peaks of hydroxyl groups were found at around 3300 cm^−1^ for all the PPL samples. However, this adsorption peak becomes weaker in the PPL−11 and PPL−11−H samples. This suggests that alkaline and high-temperature treatments can cause a decrease in phenol groups. Moreover, the peaks around 1600 cm^−1^ become broader in the range of 1600–1740 cm^−1^, which corresponds to the benzene skeleton and C=O vibrations. The enhancement of C=O stretching vibrations is possibly caused by the structural transformation between hydroxyl group and C=O. Due to the abundance of phenol groups in PPL, there were still hydroxyl groups remaining in the treated PPL, which make FT-IR spectra of PPL, PPL−11 and PPL−11−H appear similar in general.

Figure 1d shows the FT-IR spectra of various composite samples. The infrared spectrum of PPL showed the stretching vibration peak of −OH at 3294 cm^−1^. Since there is an abundance and complexity of –OH groups in PPL, the peak shows a wide and relatively strong character in the IR spectrum. The peak at 1604 cm^−1^ is the skeleton vibrations of the benzene ring in PPL. Functional groups of C=O in PPL around 1695 and 1785 cm^−1^ are also observed, indicating the multifunctional groups of PPL. The adsorption band at 1000–1300 cm^−1^ is C−O vibration, which is similar to proanthocyanidins [46]. It can be assumed that the main components of PPL extracted from larch bark are proanthocyanidins [47]. The IR spectrum of ZnO@Cel seems simpler than PPL because of the simple functional groups of cellulose. The adsorption band around 1058 cm^−1^ is C−O vibrations and the adsorption for hydroxyl groups in cellulose displays at 3336 cm^−1^, similar to that of PPL. Considering ZnO@PPL/Cel, absorption bands around 1570 and 1415 cm^−1^ are the vibrations of a benzene ring, indicating the existence of PPL in the composite. The C−O stretching vibration band is found around 1056 cm^−1^, which originates from cellulose or PPL. In addition, this peak has red shift compared with PPL and ZnO@Cel, which may be caused by the coupling effect with ZnO. Moreover, the vibrations of hydroxyl groups are relatively weaker in ZnO@PPL/Cel than in PPL, PPL−11−H and ZnO/Cel. This means that hydroxyl groups in ZnO@PPL/Cel may form hydrogen bonds with each other and weaken the vibrations. The analyses of FT-IR provide further evidence that PPL works as cross-linker which can form hydrogen bonds with both cellulose and ZnO.

### 3.2. Morphology Study

Figure 2 shows the SEM images of ZnO, ZnO/Cel and ZnO@PPL@Cel. From Figure 2a, we can see that ZnO shows the morphology of thick pellets with 1 μm in thickness and 500 nm in width. When prepared with cellulose, rod-like ZnO with the length of 1 μm is found on the surface of the cellulose. These ZnO rods have a diameter of about 200 nm and are distributed randomly. The bare cellulose still can be observed in Figure 2b. If PPL was involved during the preparation process, uniform pinecone-like ZnO was formed, which was coated with PPL. These particles are about 100–200 nm in size, which greatly reduced compared with those of ZnO and ZnO/Cel. This is evidence that the coupling effect between ZnO and PPL can restrict the size of ZnO.

Moreover, there are slices on particles, which means that pinecone-like ZnO@PPL was constructed by subunits with the size about 50 nm and form the hierarchical structure. The ZnO@PPL covered the cellulose uniformly and was well dispersed, which finally forms the composite morphology. Thus, we deduce that PPL in the composite may work as crossing linker, which connects the ZnO and cellulose. The role of PPL enhances the connection between ZnO and cellulose and results in a more uniform morphology in the composite, which can be used as active packaging material.

### 3.3. XPS and Raman Analysis

The XPS was also applied to analyse the elemental chemical circumstance of ZnO@PPL/Cel. Figure 3a is the XPS survey of ZnO@PPL/Cel and the elements C, O and Zn are found in the spectrum. Figure 3b is the spectrum of Zn2p, in which two peaks can be observed at 1021.3 and 1044.3 eV. These peaks can be assigned to the binding energies of Zn2p 3/2 and Zn2p 1/2. The deviation between these two peaks is 23 eV, indicating the existence of divalent Zn in the sample. The O1s spectrum is shown in Figure 3c. From Figure 3c one can see that three peaks can be fitted. The binding energies of the fitted peaks located at 529.9, 531.0 and 532.3 eV, respectively, which correspond to oxygen in the crystal of ZnO, and C–O–H and C–O–C bonds in both PPL and cellulose. Figure 3d is the C1s spectrum which can be fitted into 4 peaks. Three of those peaks are binding energies of C–C (284.2 eV), C–H (285.0 eV), C–O (286.0 eV), originating from both PPL and cellulose. The fourth peak located at 287.6 eV can be assigned to the C=O in PPL. The relative mass percentage of ZnO was calculated to be 65% in the composite.

To further study the structure of the composite, Raman spectra of ZnO, ZnO/Cel and ZnO@PPL/Cel were tested and are shown in Figure 4. In Figure 4, we can see that both ZnO and ZnO/Cel show the similar vibrational character: four peaks around 343, 456, 590 and 1167 cm^−1^. The peaks of 456 and 590 cm^−1^ are the first-order phonon frequencies and peaks at 345 and 1158 cm^−1^ are the second-order Raman scattering frequencies of ZnO. These vibration peaks indicate the existence of ZnO. For weak dipole symmetrical vibrations of cellulose, there is no characteristic peak of cellulose found in ZnO/cellulose. However, the Raman spectrum of ZnO@PPL/Cel shows different results: the characteristic peak of benzene in PPL is found at 1590 cm^−1^. Moreover, the peak of ZnO around 453 cm^−1^ is greatly depressed and all the peaks have a red shift of about 10 cm^−1^. These changes may be caused by the coupling effect between the PPL and ZnO, which weakens the vibrations. Raman analysis further provides the evidence that PPL can form interactions with ZnO.

### 3.4. Solid UV Diffuse Reflection and PL Spectra

Figure 5 shows the plots of the solid UV diffuse reflection of samples. It can be seen from Figure 5a that ZnO@PPL/Cel has the highest UV absorption intensity compared with ZnO and ZnO/Cel. This result means that ZnO@PPL/Cel can excite more electrons, which would improve its antibacterial activity. To explain it, the band gaps of samples were calculated by the first derivative according to solid UV diffuse reflectance data. It was found that ZnO@PPL/Cel has the narrowest band gap among ZnO, ZnO/Cel and ZnO@PPL/Cel. The narrow band gap structure allows ZnO@PPL/Cel to harvest more UV irradiation and increase its antibacterial activity.

Photo luminescence spectra further revealed the microstructure of samples and the results are shown in Figure 6a,b. It can be seen from Figure 6a,b that both ZnO and ZnO@PPL/Cel have an intensive emission at 288 nm, which is the intrinsic emission of ZnO, corresponding to the emission from conduction band to valence band. However, ZnO@PPL/Cel shows wide peak between 400 and 600 nm, which is deep centered emission. Generally, only defects in ZnO can cause this deep centered emission. This result is evidence that ZnO produces many more defects when combined with PPL and cellulose, which can active ZnO by shorting the band gap and exciting more electrons. This is consistent with the result of UV diffuse reflection analysis.

### 3.5. Antioxidant, Antibacterial Activity and Water Vapor Transmission Property

It is known that the antioxidant property of materials is related to the shelf life of food. Thus, the antioxidant activity of ZnO@PPL/Cel was evaluated by the method of DPPH radical scavenging activity [48]. As shown in Figure 7, the prepared composite has a high radical scavenging rate of 74.3% due to the presence of ZnO, which is a semiconductor [49]. However, more PPL in the composite would result in a higher radical scavenging rate of the composite. In ZnO@PPL/Cel-7, the radical scavenging rate reaches 79.6%, which is 7% higher than that of ZnO/Cel. This result indicates that PPL has a positive effect on the antioxidant property of the composite, which would prolong the shelf life of food if utilized as packaging material.

In active food package materials, antibacterial performance is very important. In our work, Gram-negative *Escherichia coli* (*E. coli*) and Gram-positive *Staphylococcus aureus* (*S. aureus*) were used to evaluate the antibacterial behavior of samples by the inhibition zone method [50]. It can be seen from Figure 8 that PPL, PPL–11 and PPL–11–H have a similar and modest antibacterial activity toward both *E. coli* and *S. aureus.* However, both ZnO/Cel and ZnO@PPL/Cel show high antibacterial activity. The diameter of the inhibition zones of ZnO/Cel against *E. coli* and *S. aureus* were 18.9 and 20.9 mm, respectively. When the sample diameter (15 mm) is subtracted, the widths of inhibition zones were 3.9 (*E. coli*) and 5.9 (*S. aureus*) mm. When coupled with PPL, ZnO@PPL/Cel samples had much better performance than ZnO/Cel: the inhibition zones for *E. coli* and *S. aureus* increased to 24.9 mm and 31.2 mm, respectively. The calculated width of inhibition zones was 5.9 and 8.1 for *E. coli* and *S. aureus*, respectively, which have been improved 150% and 137% compared to those of ZnO/Cel. These results are evidence that PPL in the composite would enhance the antibacterial activity greatly.

Some particular mechanisms increase the activity of ZnO as an antibacterial agent [51]. A common one is that ZnO can produce highly reactive oxygen species (ROS), such as superoxide, hydrogen peroxide and hydroxyl (O^2−^, H_2_O_2_ and OH^−^), which can kill bacterial cells. The production of ROS depends on the illumination light and the band gap as well as defects in the ZnO. In our case, the ZnO in composites has many more defects than ZnO and ZnO/cellulose, which would produce more ROS during application. In addition, PPL has the effects of membrane lysis, inhibition of energetic metabolism, reduction of host ligands adhesion and neutralization of bacterial toxins, which also can inhibit the growth of bacteria [52]. Thus, the special structures of ZnO and PPL in composite are the reasons for the enhanced antibacterial activity of ZnO@PPL/Cel.

To further study the role of PPL on antibacterial activity, different dosages of PPL (2.5 mL and 7 mL) were also applied to fabricate the composite. From Figure 8 and Table 1, we can observe that the dosage of PPL has little effect on its antibacterial performance. We suggest that when more PPL was loaded on ZnO, it can improve the antibacterial activity of the composite. However, more PPLs on the ZnO surface also block the light harvesting of ZnO and reduce the ROS production. Moreover, some of ZnO-produced ROS can be scavenged by PPL in the composite. These effects work together and make ZnO@PPL/Cel-5 and ZnO@PPL/Cel-7 show close antibacterial activities.

The pH sensitivity of the composite was also tested in light of its antibacterial activity. From Figure 9, it can be seen that the pH has stronger impact on *E. coli* than *S. aureus*. For *E. coli*, antibacterial activity decreases greatly with the increase of pH. When pH is 11, only a 2.4 mm inhibition zone resulted, which decreased by 52% compared with that of ZnO@PPL/Cel in the neutral conditions. However, the inhibition zone of ZnO@PPL/Cel-5 against *S. aureus* decreased by 11%. This may be caused by the protons which permeate into cells and kill the bacteria.

The water vapor transmission rate (WVTR) of samples was tested according to the reference [44]. As shown in Figure 10, one can see that the WVTA value of ZnO/Cel was about 46.12 g/(h m^2^ mm). The high value is caused by the cellulose in the composite. The combination with PPL allows the WVTR value of ZnO@PPL/Cels to decrease and fall within the range 41.75–40.01 g/(h m^2^ mm). The decrease is perhaps caused by the reduction of hydroxyl groups of cellulose, which are exploited to build hydrogen bonds with ZnO and PPL. Moreover, the high WVTR of the composite can provide a drying environment for the stored food and inhibit microbial growth.

## 4. Conclusions

Active package material based on the composite ZnO@PPL/Cel has been accessed via a simple hydrothermal method. It was found that PPL forms a coating layer on the surface of ZnO with the assistance of hydrogen bonds and restricts the ZnO growth. Thus 200 nm pine-like ZnO@PPL is observable, which is assembled from subunits of 50 nm ZnO slices. Moreover, PPL offers hydrogen bonding, which makes ZnO@PPL evenly distributed on the cellulose and together build a hierarchical structure. The narrowed band gap and massive defects were revealed in ZnO@PPL/Cel, which enhance its antibacterial properties. Antibacterial performance was evaluated against *E. coli* and *S. aureus*. The widths of inhibition zones for *E. coli* and *S. aureus*, were determined to be 5.9 mm and 8.1 mm, respectively, which were improved by 150% and 137% relative to those of ZnO/Cel. The newly-synthesized material, which has benefits including renewability, low cost, a simple synthesis method and good antibacterial activity, is promising as an active food package.

## Data Availability

Not applicable.

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
