# Peer review of "Fabrication of ZnO@Plant Polyphenols/Cellulose as Active Food Packaging and Its Enhanced Antibacterial Activity"

_ijms, 2022, doi:10.3390/ijms23095218_

Round 1
Reviewer 1 Report
The attempt made by the authors to study the potential of the ZnO@PPL/Cel fabrication against E. coli and S. aureus is highly regarded. Yet, It is a simple work that assesses only the antibacterial activity at plate level.
As mentioned in the title, the manuscript would be considered if the work reaches the level of assessing the ZnO@PPL/Cel in food packaging material.
As a reputed journal like IJMS, the data provided/ the work conducted by the authors are not sufficient and requires additional studies to validate the same.
Also, the presented MIC value in the table doesn't have SD, which shows that the result was not conducted in biological triplicate.
Reviewer 2 Report
The manuscript entitled “Fabrication of ZnO@Plant Polyphenols/Cellulose as active food package and its enhanced antibacterial activity” by Nie et al, aimed to design and develop an interesting polyphenol extract-coated ZnO nanocomposite as active package materials. The work is very interesting with promising results. I think that is worthy of publication after some revisions.
- The preparation of ZnO@PPL/Cel was performed via hydrothermal method. It is known that polyphenols are sensitive to alkaline and high temperature conditions and therefore molecules degradation can occur. Despite this, ZnO coated with PPL showed a good antibacterial activity representing a promising approach in the development of active food package. In this context I have some suggestions/questions.
- Have the authors assessed the chemical stability of their PPL extract? I think that this critical point should at least be discussed.
- The antibacterial activity of the PPL extracts (ideally before and after alkaline and hydrothermal treatment) should be assessed to compare the results with ZnO@PPL/Cel.
- The authors declare that “PPL itself can release –OH, which also can chelate bacteria, thus destroying their cell structure” (line 294). Are there any references? Are the authors sure that this effect is related only to the ability of PPL to release ROS? In fact, flavanols molecules are known to act by different mechanisms of action (such as membrane lysis, inhibition of energetic metabolism, reduction of host ligands adhesion, and neutralization of bacterial toxins) (doi.org/10.1016/j.copbio.2011.08.007; doi:10.1016/j.ijantimicag.2005.09.002). In my opinion it is impossible to assess that PPL exerts its antibacterial activity by -OH release in absence of experimental evidence. Always in this context, the authors said that “more PPL on ZnO surface, it also block the light harvesting of ZnO and reduce the ROS production” (line 301). I agree with them, but it is important to consider that polyphenol could also has radical scavenging activity and therefore some of ZnO-produced ROS can be scavenged by PPL coating at high PPL concentration.
Line 34 and 290. Please correct “CO2” in CO2” and “O2−, H2O2 and OH−“ in “O2−, H2O2 and OH−“
Line 41. Please add other references relative to anticancer effect of flavan-3-ol-based molecules (DOI: 10.3390/ijms222111833; doi.org/10.3390/molecules27030719).
Reviewer 3 Report
The authors reported work on the fabrication of ZnO/polyphenol/cellulose composite active food packaging. I have the following suggestion for the authors.
- The introduction section lacks literature on the previous studies on in situ green syntheses of ZnO on cellulose substrates. Please provide that in relation to the novelty statement of this work.
- The experimental design focuses on just one functional characterization (antibacterial characterization) other relevant properties like antioxidant, pH sensitivity, and cytocompatibility are missing.
- Material properties like porosity, pore size, air, and water permeability are all missing.
To me, the manuscript does not meet the quality of IJMS for publication.
Round 2
Reviewer 2 Report
The manuscript is now suitable for the publication.
Minor concerns:
- Line 92. “Scheme 1 the formation of ZnO@PPL/Cel.” Correct the punctuation. The same for line 173.
- Line 265. In the caption of Figure 2 the letters b, c and d are not present.
Reviewer 3 Report
The authors have revised the manuscript very well and have responded to all the queries. I recommend the publication of the article in its current form.
Author Response
Many thanks for your recommendation. You suggestions help us improve our work a lot.